# LSH-GAN enables in-silico generation of cells for small sample high dimensional scRNA-seq data

Snehalika Lall[1], Sumanta Ray [iD] [2,3✉] & Sanghamitra Bandyopadhyay[1✉]

A fundamental problem of downstream analysis of scRNA-seq data is the unavailability of enough cell samples compare to the feature size. This is mostly due to the budgetary constraint of single cell experiments or simply because of the small number of available patient samples. Here, we present an improved version of generative adversarial network (GAN) called LSH-GAN to address this issue by producing new realistic cell samples. We update the training procedure of the generator of GAN using locality sensitive hashing which speeds up the sample generation, thus maintains the feasibility of applying the standard procedures of downstream analysis. LSH-GAN outperforms the benchmarks for realistic generation of quality cell samples. Experimental results show that generated samples of LSH-GAN improves the performance of the downstream analysis such as feature (gene) selection and cell clustering. Overall, LSH-GAN therefore addressed the key challenges of small sample scRNA-seq data analysis.

[1] Machine Intelligence Unit, Indian Statistical Institute, Kolkata, West Bengal 700108, India. [2] Health Analytics Network, Pittsburgh, PA, USA. [3] Department of Computer Science and Engineering, Aliah University, Kolkata, India. ✉email: sumanta.ray@aliah.ac.in; sanghami@isical.ac.in

Recently, the emergence of high dimensional biological data such as single-cell RNA sequence (scRNA-seq) data has posed challenges to machine-learning researchers[1,2]. The high dimension, and small-sample size (HDSS) data handling is difficult for downstream analysis particularly for feature selection (FS). It affects later stages of downstream analysis such as cell clustering, marker selection, and annotation of cell clusters. A few outliers can drastically affect the FS techniques, and the selected feature sets may not be adequate to discriminate the classes[3]. Moreover, high dimensionality increases the computational time beyond acceptability.

High dimensional small-sample (HDSS) data is prevalent in the single-cell domain due to the budgetary constraint, ethical consideration of single-cell experiments, or simply because of the small number of available patient samples. Whatever the reason is, too few observations (cell sample) in the single-cell data may create problems in the downstream analysis. This is because a small-sample size may not reflect the whole population accurately, which surely degrades the performance of any model. The general pipeline of scRNA-seq downstream analysis starts with preprocessing (normalization, quality control) of the raw count matrix and then going through several steps which include identification of relevant genes, clustering of cells, and annotating cell clusters with marker genes[4–8]. Each step has a profound effect on the next stage of analysis. The gene selection step identifies the most relevant features/genes from the normalized/preprocessed data and has an immense impact on cell clustering[9,10]. The general procedure for selecting relevant genes which are primarily based on high variation (highly variable genes)[11,12] or significantly high expression (highly expressed genes)[4] suffers from a small-sample effect. The general FS techniques also failed to provide a stable and predictive feature set in this case due to an ultra-large size of feature (gene). One way to solve this issue is to go for a robust and stable technique that does not overfit the data. A few attempts[9,13,14] were observed recently which embed statistical and information-theoretic approaches. Although these methods result in stable features, however, these are not performed well in small-sample scRNA-seq data.

Recently computational researchers are gaining interest in this field. Some methods like cscGAN[15], Splatter[16], SUGAR[17] are already developed which uses different techniques (like generative model, statistical framework) to successfully simulate the samples of specific cell types or subpopulations. The challenge in this task is to handle the sparsity and heterogeneity of the cell populations which define the specific characteristics of scRNA-seq data. In this paper, we propose a generative model to sort out this problem in HDSS scRNA-seq data. We use generative adversarial model to generate realistic cell samples from a small number of available samples of HDSS scRNA-seq data. Generative adversarial network (GAN)[18–21] has already been shown to be a powerful technique for learning and generating complex distributions[22,23]. However, the training procedure of GAN is difficult and unstable. The training suffers from instability because both the generator and the discriminator model are trained simultaneously in a game that requires a Nash equilibrium to complete the procedure. Gradient descent does this, but sometimes it does not, which results in a costly time-consuming training procedure. The main contribution here is in modifying the generator input that results in a fast training procedure. We create a subsample of original data based on locality-sensitive hashing (LSH) technique and augment this with noise distribution, which is given as input to the generator. Thus, the generator does not take pure noise as input, instead, we introduce a bias in it by augmenting a subsample of data with the noise distribution.

Researchers are still trying to find improved versions of the GAN to use in different domains. Most of the variations such as progressive GAN (PGAN)[24], Wasserstein GAN (WGAN)[22] try to train the model quicker than the conventional GAN. Unlike PGAN and WGAN, conditional GAN (CGAN)[25] operates by conditioning the conventional model on additional data sources (maybe class label or data from different modalities) to dictate the data generation. In our model, we direct our attention to the additional sample generation from HDSS data. However, the generated sample size becomes increasingly large with more features, the generation of which may not be feasible for conventional generative models. Augmenting subsample of real data distribution ($p_{data}(x)$) with the prior noise ($p_z(z)$) makes the training procedure of our model faster than the conventional GAN. We theoretically proved that the global minimum value of the virtual training criterion of the generator is less than the traditional GAN ($<-\log 4$).

Here, we provide the following: (i) The proposed model address the problem of downstream analysis (particularly gene selection and clustering) on HDSS scRNA-seq data. (ii) LSH-GAN is able to generate realistic samples in a faster way than the traditional GAN. This makes LSH-GAN more feasible to use in the feature (gene) selection problem of scRNA-seq data. (iii) LSH-GAN can produce more realistic cell samples than the other existing benchmarks. (iv) Here we derive a training procedure of generator that combines subsamples of original data with pure noise and takes this as input. (v) For a fixed number of iteration LSH-GAN performed better than the traditional GAN in generating realistic samples. (vi) Gene selection and clustering on the generated samples of LSH-GAN provide excellent results for small-sample and large-feature sized single-cell data.

## Results and discussion

In the following, we first describe the workflow of our analysis pipeline. Next, experimental validation of the proposed model is carried out by comparing it with several state-of-the-arts in real-life scRNA-seq data. Finally, we used LSH-GAN to select genes from HDSS scRNA-seq data. We use benchmark gene selection techniques on the generated samples and used one single-cell clustering technique to validate the selected genes.

**Proposed model: LSH-GAN**. Figure 1 describes the workflow of our analysis pipeline. Figure 1a describes the application of the proposed LSH-GAN model in the feature selection task of the HDSS scRNA-seq data, while Fig. 1b depicts basic building blocks of the model. The following subsections describe in brief.

*LSH step: sampling of input data.* Locality-sensitive hashing (LSH)[14,26,27] is widely used in nearest neighbor searching to reduce the dimensionality of data. LSH utilizes locality-sensitive hash functions which hash similar objects into the same bucket with a high probability. The number of buckets is much lesser than the universe of possible items, thus reduces the search space of the query objects (see Supplementary Note 1 for a detailed description of LSH technique). The intuition behind LSH step is to capture non-redundant and widely separated samples from the original scRNA-seq data, which helps to learn the complex distribution of the data in a holistic way. Please note that the aim of LSH step is to provide a prior sense of information about real data distribution to the generator network.

In this work first, the unique hash codes which depict the local regions or neighborhoods of each data point are produced. For this, we utilized python sklearn implementation of *LSHForest* module with default parameters.

An approximate neighborhood graph ($k$-nn graph) is constructed by using $k = 5$ for each data point. This step computes the euclidean distances between the query point and its candidate

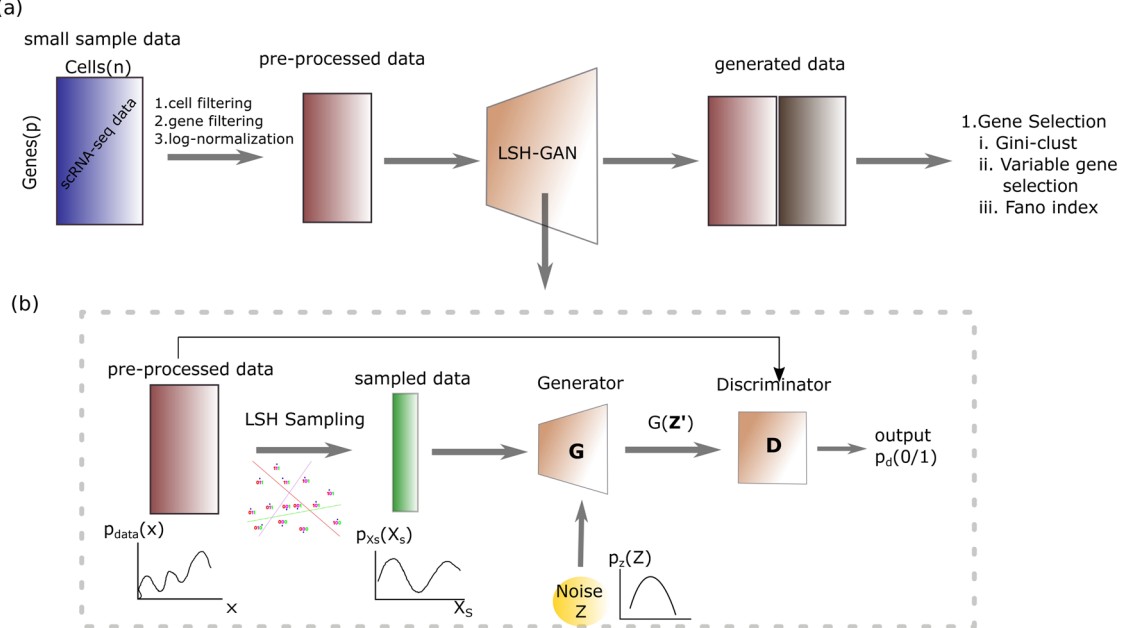

**Fig. 1 Workflow of LSH-GAN. a** Gene selection task in HDSS scRNA-seq data using generated samples with LSH-GAN model. **b** Detail architecture of LSH-GAN.

neighbors. Sampling is carried out in a 'greedy' fashion where each data point is traversed sequentially and its corresponding five nearest neighbors are flagged out which never visited again. Thus after one traversing a sub-set of samples is obtained which is further down-sampled by performing the same step iteratively.

*Generator of LSH-GAN.* The generator function ($G$) is modified by augmenting its taken input data. Instead of giving the pure noise ($p_z(z)$) as input we augment a subsample of real data distribution ($p_{data}(x)$) with it. The sampling of the input data is done in the LSH step. Thus the generator ($G$) function builds a mapping function from $\widehat{z}$ to data space ($x$) as $G(\widehat{z}; \theta_g)$ and is defined as: $G(.) : \widehat{z} \rightarrow x$. Modifying the generator in this way we claim that it can increase the probability of generating samples of real data in lesser time.

*Discriminator of LSH-GAN.* Here discriminator ($D$) takes both the real data $p_{data}(x)$ and generated data coming from generator ($G(\widehat{z})$), with probability density ($p_{\widehat{z}}(\widehat{z})$) and returns the scalar value, $D(x)$ that represents the probability that the data $x$ is coming from the real data: $D(.) : x \rightarrow [0, 1]$.

So, the value function can be written as:

$$L(D, G) = \min_G \max_D (E_{x \sim p_{data}(x)} \log(D(x)) + E_{\widehat{z} \sim p_{\widehat{z}}(\widehat{z})} \log(1 - D(G(\widehat{z}))))$$

(1)

$D$ and $G$ form a two-player minimax game with value function $L(G, D)$. We train $D$ to maximize the probability of correctly validating real data and generated data. We simultaneously train $G$ to minimize $\log(1 - D(G(\widehat{z})))$, where $G(\widehat{z})$ represents the generated data from the generator by taking the noise ($p_z$) and the sampled data $p_{x_s}(x_s)$ as input.

*Feature/gene selection using LSH-GAN.* The generated cell samples are utilized for the gene selection task. We have employed five well-known gene selection methods (with default parameters) of scRNA-seq data adopted for validation: *GLM-PCA*[28], *CV²Index*, M3Drop[29], *Fano Factor*[30], and Highly Variable Gene (HVG) selection of Seurat V4[31]. Single-cell clustering method

(SC3) technique is utilized to validate the selected genes from the generated samples.

The whole algorithm and the sampling procedure are described in Table 1.

**Experimental settings.** The number of nearest neighbor ($k$) and the number of iteration ($t$) are two main parameters of the LSH step (see Table 1), tuning of which affects the amount of sampling given to the generator for training the LSH-GAN model. We vary $k$ and $t$ in the range $\{5, 10, 15, 20\}$ and $\{1, 2\}$, respectively, and choose that value for which the Wasserstein distance[22] between generated and real samples are reported to be minimum. We fixed the amount of sampling using $k = 5, t = 1$ for Pollen, Yan, Darmanis datasets and $k = 5, t = 2$ for Klein dataset and Melanoma datasets (see Supplementary Table 1). Similarly, we choose the epoch ($e_{opt}$), which results in the lowest Wasserstein metric. For example, we take $e_{opt}$ as 10k, 30k, 10k, 15k, and 25k for the dataset Darmanis, Yan, Pollen, Klein, and Melanoma, respectively (see Supplementary Fig. 2). For generating hash code from LSH sampling, *LSHForest* of *scikit-learn* version 0.19.2 is utilized.

We take the adaptive learning rate optimization algorithm implemented in ADAM optimizer in python Tensorflow version 1.9.2. Generator ($G$) and discriminator ($D$) uses 2-layer multi-layer perceptrons with hidden layer dimension as (16, 16). For traditional GAN, we retain the same settings as LSH-GAN for G and D networks.

For benchmarking our method we have utilized three state-of-the-art techniques widely used for sample generation: cscGAN[15], SUGAR[17], and Splatter[16]. For these three methods, We adopted the code (with default parameters) provided on the Github page of the original publications.

Five well-known gene selection methods (with default parameters) of scRNA-seq data are adopted for validation: *GLM-PCA*[28], *CV² Index*, M3Drop[29], *Fano Factor*[30], and Highly Variable Gene (HVG) selection of Seurat V4[31]. We select the top 500 features (genes) using all three feature selection methods on scRNA-seq datasets. For validation purposes, Wasserstein metric[22] is utilized to estimate the quality of the generated data. Clustering of scRNA-seq data is performed using *SC3*[32] technique with default parameters. Clustering performance is

**Table 1 LSH-GAN algorithm.**

**Input:**    Data Matrix (**x**), number of Training iterations, number of nearest-neighbor (**k**), number of iterations for sub-sampling (**t**)
**Output:**    Generated data (**G$_{out}$**).
1:    **for** number of training iterations **do**
2:        $x_s$ = LSH-SAMPLING($x,k,$t)
3:        augment $p_{x_s}(x_s)$ with prior noise $p_z(z)$ and give this ($p_{\hat{z}}(\hat{z})$) to the generator, $G$.
4:        real data $p_{data}(x)$ and generated data $p_g(x)$ is given to discriminator $D$.
        **Update the Discriminator**, $D$
5:        $\Delta_d = \sum_{i=1}^{n} \log(D(x_i)) + \log(1 - D(G(\hat{z})_i)))$
        **Update the Generator**, $G$
6:        $\Delta_g = \sum_{i=1}^{n} \log(1 - D(G(\hat{z})_i)))$
7:        **end for**
        {The adaptive momentum gradient decent rule is used in our experiment.}
8:    **procedure** LSH sampling($x, k, t$)
9:    Execute Locality Sensitive Hashing (LSH) on $x$ and prepare a $k$-Nearest Neighbor list for each data point.
10:    **for** number of iteration of sub-sampling $t$ **do**
11:        visit each data point sequentially in the order as it appears in data.
12:        if the data point is not visited earlier, select the data point and discard all its $k$ neighbors from its nearest-neighbor list.
13:    **end for**
14:    **end procedure**

---

**Table 2 Wasserstein distance between generated and real data distribution.**

| Nearest neighbor | Model | Epoch | | | |
|---|---|---|---|---|---|
| | | 10,000 | 15,000 | 20,000 | 25,000 |
| | | Wassertein distance | | | |
| $k = 5$ | LSH-GAN | **0.46** | **0.35** | **0.33** | **0.45** |
| $k = 10$ | LSH-GAN | 1.09 | 0.89 | 0.83 | 0.82 |
| $k = 15$ | LSH-GAN | 1.36 | 0.89 | 1.45 | 0.87 |
| $k = 20$ | LSH-GAN | 1.53 | 1.35 | 1.19 | 0.83 |
| | GAN | 1.71 | 1.73 | 1.75 | 1.70 |

Model is trained on synthetic data of size 100 × 1000 Gaussian mixture data with two non-overlapping classes.
The minimum distances are represented as bold face.

evaluated using the adjusted Rand index (ARI), normalized mutual information (NMI).

**LSH-GAN improves performance of traditional GAN on simulated data.** First, we train the LSH-GAN on HDSS synthetic data and generate realistic samples to compare against the traditional GAN model. For this, we create a 2-class non-overlapping Gaussian mixture data consisting 100 samples and 1000 features by taking the mean ($\mu$) of the data in the range of 5 to 15 for class-1 and $-15$ to $-5$ for class-2. The covariance matrix ($\Sigma$) is taken for all the samples using the formula $\Sigma = (\rho^{|i-j|})$, where $i$, $j$ are row and column index, and $\rho$ is equal to 0.5. We calculate Wasserstein metric to estimate the quality of the generated data. The Wasserstein distance between the real data distribution ($p_{data}$) and the generated data distribution ($p_g$) to estimate the quality of the generated data. We use different settings of $k$th ($k = 5, 10, 15, 20$) nearest neighbor to generate subsample of data from LSH sampling procedure. In each case, the sampled data ($p_{x_s}$) is augmented with prior noise ($p_z$) and given to the generator of LSH-GAN for model training.

For comparison with the traditional GAN model, we use the data with train: test split of 80:20 and calculate the Wasserstein metric between the test sample and the generated sample. Table 2 shows the values of the metric for LSH-GAN and traditional GAN model in different range of epochs and nearest neighbors $k$. A closer look into Table 2 reveals that the performance of LSH-GAN (at 10,000 epoch and $k = 5$) is far better than the traditional

GAN model with 25,000 epochs. Notably, for less amount of sampling (larger $k$), LSH-GAN needs more iterations for training. As for particular example, the performance of LSH-GAN achieved on 20,000 epoch and $k = 20$ is rivaled only at 10,000 epoch for $k = 10$. Thus it is evident from the results that reducing the amount of sampling needs more epochs and thus needs more training time for the LSH-GAN model to converge. Figure 2 also supports this statement. Here, the two models (LSH-GAN, and traditional GAN) are trained to simulate a two-dimensional synthetic data of known distribution, for which the LSH-GAN can able to generate samples that are more real than the traditional GAN, in a lesser number of iteration.

**Comparison of LSH-GAN with benchmarks in HDSS scRNA-seq data.** We compare LSH-GAN with four existing benchmarks: cscGAN[15], splatter[16], SUGAR[17], traditional GAN[18], and its two variants f-GAN[23] and w-GAN[33]. Since the evaluation of the generative model is notoriously difficult[34,35], we first use Wasserstein distance to compare the real data distribution and generated data distribution coming from different competing models. We also used UMAP visualization, and marker genes expression to visualize the generated cell samples. We create UMAP visualization for LSH-GAN, SUGAR, and cscGAN, as these methods are more associated with the generation of single-cell data. Figure 3a–c shows the two-dimensional UMAP representation of generated and real cell samples from the test data for four competing models. Melanoma data is utilized for this experiment. As can be seen from this figure, LSH-GAN can able to retain the distribution of the original cell samples. This can also be supported by the Wasserstein distance (see Fig. 3f) measured between real data and generated data distribution. To know how the expression of the marker genes is retained in the generated data, we plot the expression of marker gene CD8A (marker for CD8 T cell) and MS4A1 (marker for B cell) in the two-dimensional UMAP space for both the real and generated samples of LSH-GAN (see Fig. 3d, e). It reveals from the figure that marker genes CD8A and MS4A1 show similar expression patterns (high expression) both in real and generated cell samples.

We also validate the generated samples by training a classifier (random forest) to see whether it can able to distinguish the samples coming from two different distributions (real and generated). The aim is to see whether the model can discriminate between the real and generated cell samples accurately. Table 3

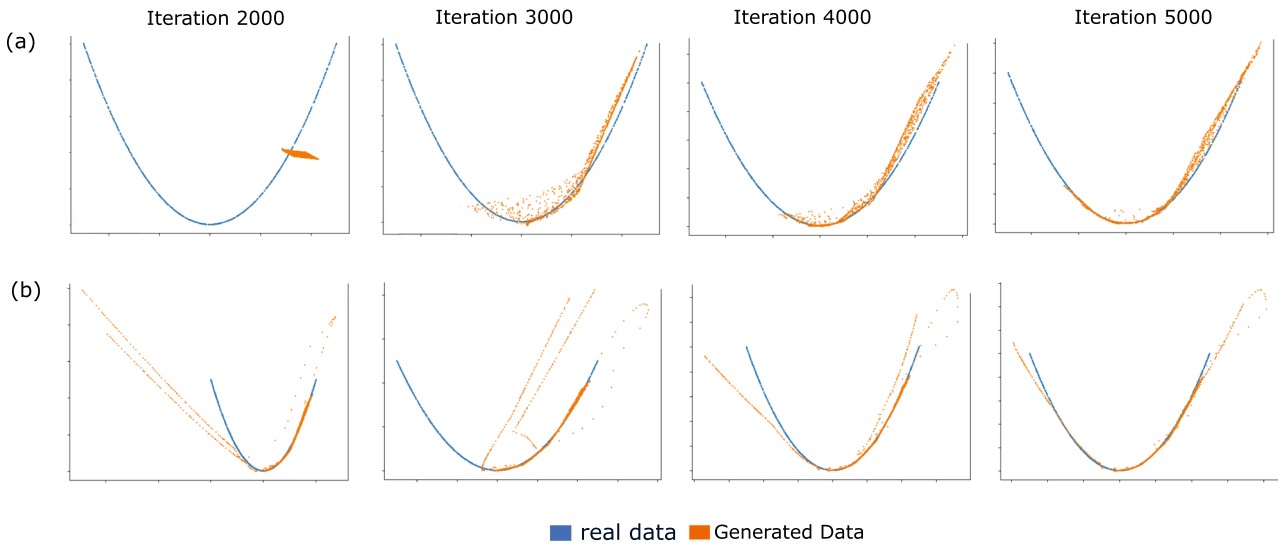

**Fig. 2 A toy example demonstrating generation of a two dimensional data of known distribution.** Results show the distribution of generated data and real data for traditional GAN (upper row, **a**) and LSH-GAN (lower row, **b**).

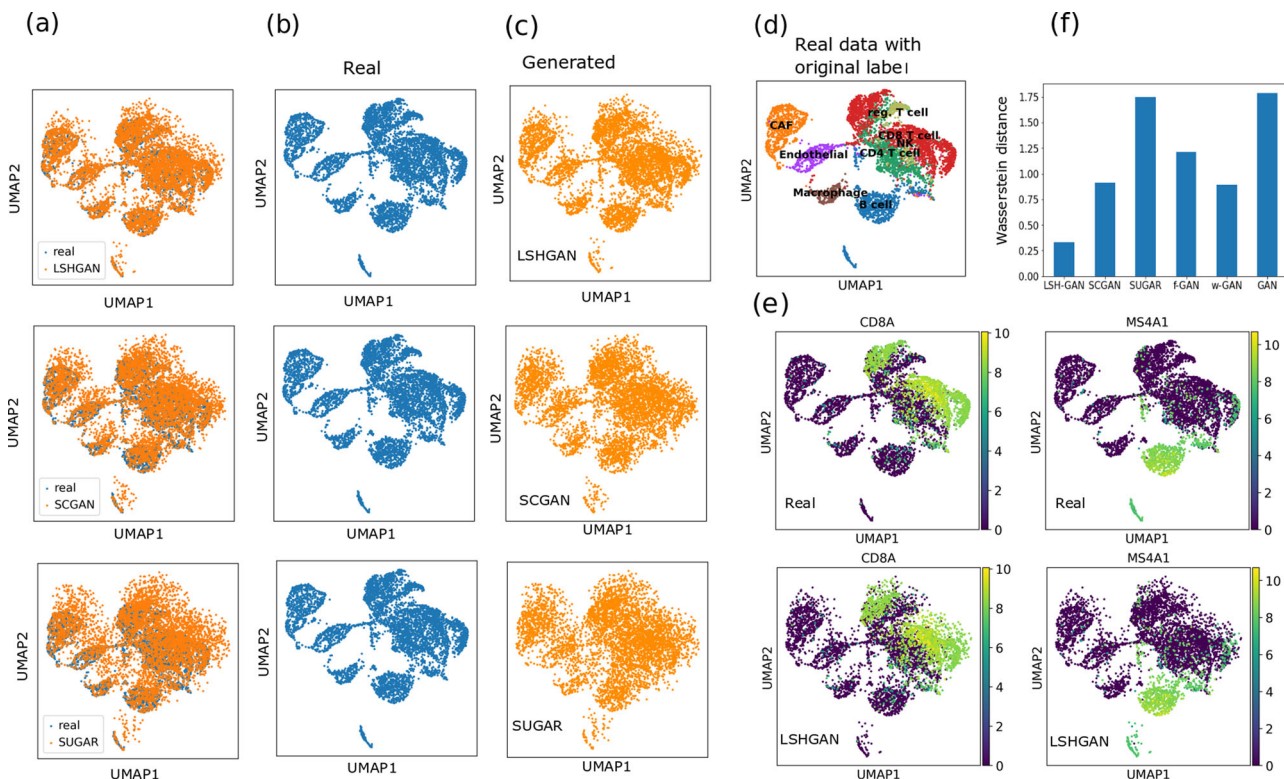

**Fig. 3 Comparisons of LSH-GAN with the state-of-the-arts on the melanoma data. a–c** UMAP visualization of real and generated cell samples of melanoma data. **d** UMAP visualization of real scRNA-seq data with the original labels. **e** Expression values (shown in color bar) of two marker genes *CD8A* (marker of CD8 T cell) and *MS4A1* (marker of B cell) in real and generated data. **f** Barplot describing the Wasserstein distance between the generated and real cell sample.

shows the cross-validation AUC score of the random forest classifier for five scRNA-seq datasets. It reveals from the table that for LSH-GAN, the AUC scores hardly reach 0.6 (only for melanoma data) suggesting a chance-level performance of RF model. This suggests the generated data obtained from LSH-GAN is highly similar to the real data.

**Gene selection from generated sample of HDSS scRNA-seq data**. Here, we aim to address the problem of gene selection in

HDSS scRNA-seq data using the generated samples. We augment the generated sample with original data to make the sample to feature ratio as 1.5. The augmented data is utilized for gene selection. Here, we have employed five feature selection methods (Highly Variable Gene (HVG) selection of Seurat V3/V4, M3Drop, GLM-PCA, and *Fano Factor*, $CV^2$-index), widely used for the gene selection task in scRNA-seq data and one single-cell clustering method (SC3) technique to validate the selected genes from the augmented data.

**Table 3 Table shows results of applying random forest classifier for discriminating real and generated samples coming from different competing methods.**

| | AUC Score | | | | |
| | Yan | Darmanis | Pollen | Klein | Melanoma |
|---|---|---|---|---|---|
| cscGAN | 0.65 ± 0.02 | 0.68 ± 0.01 | 0.64 ± 0.02 | 0.62 ± 0.02 | 0.66 ± 0.01 |
| Splatter | 0.69 ± 0.01 | 0.69 ± 0.02 | 0.67 ± 0.03 | 0.65 ± 0.01 | 0.72 ± 0.02 |
| SUGAR | 0.67 ± 0.02 | 0.66 ± 0.03 | 0.61 ± 0.02 | 0.64 ± 0.02 | 0.68 ± 0.01 |
| GAN | 0.72 ± 0.02 | 0.71 ± 0.02 | 0.73 ± 0.02 | 0.72 ± 0.02 | 0.76 ± 0.03 |
| f-GAN | 0.69 ± 0.01 | 0.70 ± 0.02 | 0.63 ± 0.02 | 0.62 ± 0.02 | 0.61 ± 0.03 |
| w-GAN | 0.61 ± 0.01 | 0.67 ± 0.03 | 0.63 ± 0.02 | 0.61 ± 0.02 | 0.64 ± 0.02 |
| LSH-GAN | **0.59 ± 0.01** | **0.60 ± 0.02** | **0.58 ± 0.02** | **0.57 ± 0.01** | **0.60 ± 0.01** |

The average AUC score (with 5-fold cross-validation) is reported for each dataset.
The lowest AUC scores are highlighted with bold face.

**Table 4 The table shows adjusted Rand index (ARI), and normalized mutual information (NMI) scores of clustering results on real-life scRNA-seq data.**

| Dataset | FS Method | Clustering results on scRNA-seq data | | | | | | | | | | | |
| | | using features from combined data | | | | | | | | | | using features from original data | |
| | | LSH-GAN | | SUGAR | | cscGAN | | Splatter | | GAN | | | |
| | | ARI | NMI | ARI | NMI | ARI | NMI | ARI | NMI | ARI | NMI | ARI | NMI |
|---|---|---|---|---|---|---|---|---|---|---|---|---|---|
| Darmanis | GLM-PCA | 0.634 | 0.66 | 0.413 | 0.43 | 0.531 | 0.54 | 0.42 | 0.43 | 0.129 | 0.15 | 0.4 | 0.41 |
| | Fano Factor | 0.535 | 0.54 | 0.319 | 0.328 | 0.457 | 0.467 | 0.38 | 0.4 | 0.27 | 0.28 | 0.34 | 0.36 |
| | CV2 Index | 0.598 | 0.61 | 0.42 | 0.453 | 0.51 | 0.53 | 0.481 | 0.51 | 0.461 | 0.48 | 0.457 | 0.462 |
| | M3Drop | 0.648 | 0.665 | 0.513 | 0.537 | 0.58 | 0.59 | 0.507 | 0.52 | 0.48 | 0.512 | 0.46 | 0.48 |
| | HVG (Seurat V4) | 0.68 | 0.702 | 0.51 | 0.54 | 0.556 | 0.573 | 0.539 | 0.54 | 0.46 | 0.472 | 0.43 | 0.427 |
| Yan | GLM-PCA | 0.895 | 0.9 | 0.709 | 0.713 | 0.798 | 0.8 | 0.715 | 0.72 | 0.62 | 0.63 | 0.66 | 0.678 |
| | Fano Factor | 0.821 | 0.843 | 0.79 | 0.8 | 0.801 | 0.81 | 0.768 | 0.77 | 0.73 | 0.75 | 0.713 | 0.72 |
| | CV2 Index | 0.891 | 0.913 | 0.801 | 0.812 | 0.825 | 0.84 | 0.793 | 0.81 | 0.719 | 0.743 | 0.7 | 0.723 |
| | M3Drop | 0.898 | 0.904 | 0.802 | 0.82 | 0.796 | 0.81 | 0.79 | 0.823 | 0.761 | 0.783 | 0.71 | 0.732 |
| | HVG (Seurat V4) | 0.91 | 0.917 | 0.811 | 0.82 | 0.891 | 0.9 | 0.802 | 0.81 | 0.81 | 0.83 | 0.8 | 0.81 |
| Pollen | GLM-PCA | 0.835 | 0.82 | 0.78 | 0.77 | 0.819 | 0.8 | 0.793 | 0.8 | 0.788 | 0.77 | 0.78 | 0.76 |
| | Fano Factor | 0.933 | 0.913 | 0.878 | 0.86 | 0.916 | 0.88 | 0.88 | 0.87 | 0.815 | 0.8 | 0.712 | 0.7 |
| | CV2 Index | 0.94 | 0.921 | 0.906 | 0.88 | 0.908 | 0.89 | 0.89 | 0.86 | 0.831 | 0.81 | 0.81 | 0.8 |
| | M3Drop | 0.918 | 0.9 | 0.864 | 0.854 | 0.897 | 0.87 | 0.79 | 0.77 | 0.758 | 0.74 | 0.735 | 0.723 |
| | HVG (Seurat V4) | 0.958 | 0.93 | 0.916 | 0.9 | 0.897 | 0.876 | 0.868 | 0.85 | 0.801 | 0.79 | 0.82 | 0.81 |
| Klein | GLM-PCA | 0.815 | 0.79 | 0.769 | 0.75 | 0.784 | 0.76 | 0.731 | 0.71 | 0.581 | 0.57 | 0.66 | 0.64 |
| | Fano Factor | 0.8 | 0.78 | 0.742 | 0.72 | 0.782 | 0.761 | 0.77 | 0.76 | 0.699 | 0.66 | 0.796 | 0.77 |
| | CV2 Index | 0.82 | 0.79 | 0.71 | 0.7 | 0.761 | 0.75 | 0.709 | 0.69 | 0.69 | 0.67 | 0.68 | 0.65 |
| | M3Drop | 0.837 | 0.824 | 0.794 | 0.77 | 0.769 | 0.74 | 0.718 | 0.7 | 0.61 | 0.6 | 0.607 | 0.59 |
| | HVG (Seurat V4) | 0.898 | 0.86 | 0.861 | 0.84 | 0.857 | 0.83 | 0.785 | 0.77 | 0.73 | 0.71 | 0.739 | 0.71 |

Data generated by the five competing methods are utilized for gene selection. Five gene selection methods are utilized to find out the most variable genes, which are further used for clustering of original scRNA-seq data. The last column represents the clustering results using the selected features from the original scRNA-seq data.

LSH-GAN is compared with five other state-of-the-arts in four HDSS scRNA-seq datasets (Darmanis, Yan, Pollen, and Klein datasets). We exclude Melanoma data for this analysis as it already has larger sample size compared to the feature size (sample:feature is 3.46). The aim is to know whether the selected features/genes from the generated combined data can lead to a pure clustering of cells. Table 4 shows the comparisons of the ARI and NMI values resulting from the cell clustering. It is evident from the table that features/genes selected from the generated combined data of the LSH-GAN model produce better clustering results than the other competing models. The last column of Table 4 shows the ARI and NMI scores of clustering results with the selected features (genes) from original scRNA-seq data.

**Selected genes using LSH-GAN can effectively predict cell clusters**. Here our aim is to investigate whether the selected genes from generated scRNA-seq data are effective for cell clustering. We have utilized the generated data from the four scRNA-seq datasets for gene selection. A widely used single-cell clustering method SC3[32] is adopted for cell clustering. Figure 4a depicts the t-SNE visualization of predicted clusters and their original labels

for Yan and Pollen datasets (see Supplementary Note 3 and Supplementary Fig. 1 for the results of the other two datasets). Figure 4b represents heatmaps of cell × cell consensus matrix. Each heatmap signifies the number of times a pair of cells is appearing in the same cluster[32]. Here two cells are said to be in different clusters if the score is zero (blue color). Similarly, a score '1' (red) signifies two cells that belong to the same class. Thus completely red diagonals and blue off-diagonals represent a perfect clustering. A careful notice on Fig. 4a and b reveals a perfect match between the original and predicted labels for YAN and Pollen datasets.

**LSH-GAN is robust for data with different batches**. To know how the generated data of LSH-GAN is affected by the data of different batches, we performed this analysis. We first download the processed datasets from a github repository (https://github.com/JinmiaoChenLab/Batch-effect-removal-benchmarking) of Tran et al.[36]. The data consists of human blood dendritic cell (DC) cells created with Smart-Seq2 technology in two different batches. Both batches contain 96 pDC and 96 double negative cells. Each batch has one biologically similar unshared cell type:

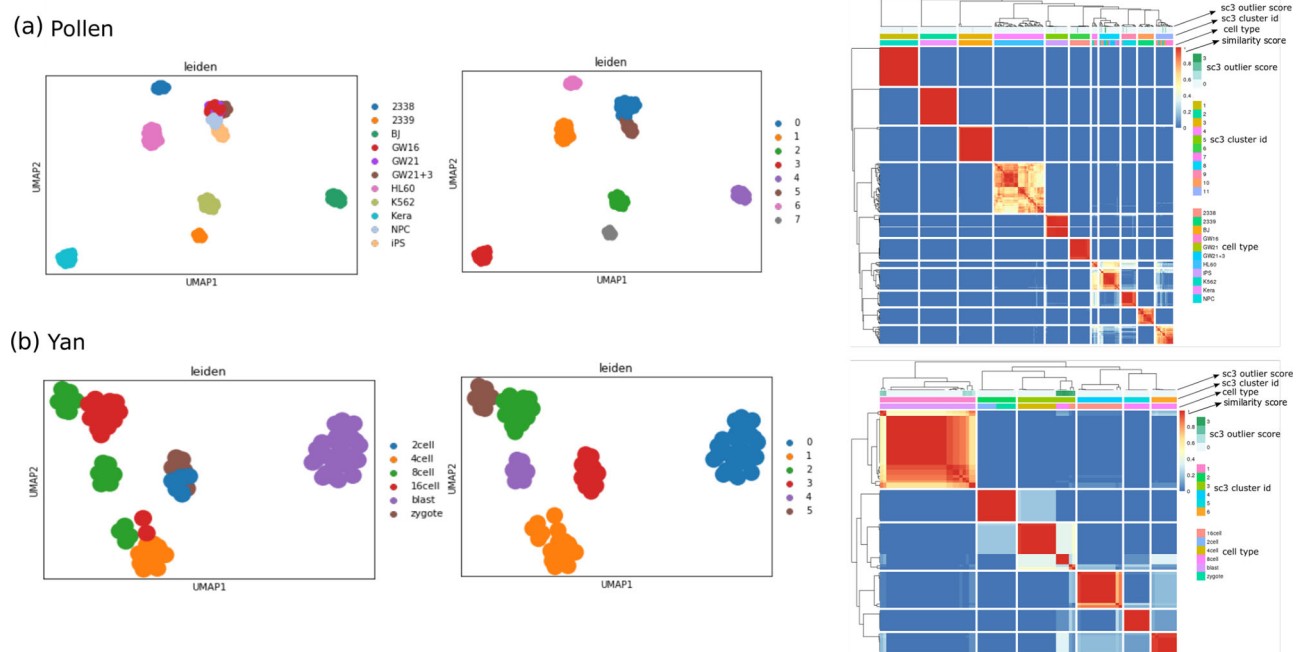

**Fig. 4 Clustering results of Pollen and Yan datasets. a** Two-dimensional UMAP visualization of clustering results (original and predicted labels). **b** Consensus clustering plots of obtained clusters.

CD141 cells in the first batch and CD1C cells in the second batch. The two batches contain a total of 288 cells over 16,594 genes. We have applied LSH-GAN on the first (or second) batch and select genes from the generated data. These genes are utilized to cluster the second (or first) batch. In both cases, we got high ARI scores (0.834 and 0.891 for two cases, respectively). For gene selection, we have utilized HVG of Seurat v4.

**Conclusions**. In this paper, we present a novel and faster way of generating cell samples of HDSS single-cell RNA-seq data using a generative model called LSH-GAN. We update the training procedure of GAN using locality-sensitive hashing which can produce realistic samples in a lesser number of iteration than the traditional GAN model. We utilized the generated data in the standard procedure of downstream analysis for analyzing real-life scRNA-seq data. Particularly, we demonstrated that the recent and benchmark approaches of gene selection and cell clustering produce excellent results on the generated cell samples of LSH-GAN. Our preliminary simulation experiment also suggests that for fixed number of training iteration the proposed model can generate more realistic samples than the traditional GAN model. This observation is also established theoretically by proving that the cost of value function is less than $-\log 4$ which is the cost for traditional GAN at the global minimum of virtual training criterion ($p_g = p_{data}$).

We demonstrated that the generated samples of LSH-GAN are useful for gene selection and cell clustering in HDSS scRNA-seq data. particularly the excellent results of LSH-GAN over the recent benchmark methods support its usability for generating realistic cell samples. For validation of the generated cell samples, we use the conventional steps of downstream analysis for scRNA-seq data. We employ five widely used gene selection techniques and one single-cell clustering technique for gene selection and grouping of cells. The precise clustering of cells demonstrates the quality of generated cell samples using the LSH-GAN model.

One limitation of our method is that for feature selection we hardly found any linear relationship between the clustering

results with the sample size of generated scRNA-seq data. The correct sample size should be selected by using a different range of values between $0.25p$ to $1.5p$, where $p$ is the feature size. There may be some effects of different parameters related to single-cell clustering (SC3 method) and feature selection (e.g., different FS methods, number of selected features, etc.) which may play a critical role in the clustering performance. However, we found clustering results are always better for the generated data with more than $1p$ ($p$ is the feature size) sample size. This observation suggests that for feature selection in HDSS data, whenever we produce samples larger than the feature size we will end up with a better clustering. The feasibility of generating such samples is justified by the faster training procedure of LSH-GAN model.

It may interesting to speculate how well LSH-GAN can be useful for generating data of other domains, such as for image analysis, bulk RNA-seq analysis, and spatial transcriptomics. For bulk RNA-seq data, one can directly apply LSH-GAN, keeping the same setup of LSH-based sampling procedure (see Supplementary Note 4 for one such analysis). For image data, the LSH-based sampling procedure needs to be further developed, so that it can be useful to capture non-redundant images (a subsample of image) from the whole datasets. For spatial transcriptomic domain, the obtained data from the technology has spatial arrangement of cell types within a tissue and thus extremely useful to understand normal development and disease pathology. In-silico generation of this data may find great interest to the machine learning researcher as the model should capture the location-wise heterogeneity of the real samples.

Taken together, the proposed model can generate good quality cell samples from HDSS scRNA-seq data in a lesser number of iteration than the traditional GAN model. Results show that LSH-GAN not only leads over the benchmarks in the cell sample generation of scRNA-seq data but also accelerates the way of gene selection and cell clustering in the downstream analysis. We believe that LSH-GAN may be an important tool for computational biologists to explore the realistic cell samples of HDSS scRNA-seq data and its application in downstream analysis.

**Table 5 A brief summary of the datasets used in the experiments.**

| # Serial | Dataset name | Features | Instances | Class |
|---|---|---|---|---|
| 1 | Yan[40] | 20,214 | 90 | 7 |
| 2 | Klein[41] | 24,175 | 2717 | 4 |
| 3 | Darmanis[39] | 22,088 | 466 | 9 |
| 4 | Pollen[38] | 23,794 | 299 | 11 |
| 5 | Melanoma[37] | 19,783 | 68,579 | 14 |

## Methods

**Overview of datasets**. We have used five public benchmark scRNA-seq datasets: Melanoma[37], Pollen[38], Darmanis[39], Yan[40], and Klein[41] downloaded from Gene Expression Omnibus (GEO) https://www.ncbi.nlm.nih.gov/geo/. Table 5 shows a detailed summary of the used datasets (see Supplementary Note 2 for description). The sample:feature ratio for all the datasets except Melanoma is <0.012. For Melanoma the ratio is quite large (3.41). We retain this data to know the efficacy of our model in both small and large sample data.

**Data preprocessing**. The raw count matrix $M \in \mathcal{R}^{c \times g}$, where $c$ and $g$ represent the number of cells and genes, respectively, is normalized using *Linnorm*[42] Bioconductor package of R. We select cells having more than a thousand expressed genes (non zero values) and choose genes having a minimum read count more than 5 in at least 10% of the cells. $\log_2$ normalization is performed on the transformed matrix by adding one as a pseudo count.

**Formal details of LSH-GAN**. In this section, we first provide a short description of GAN and then explain the theoretical foundation of LSH-GAN model.

*Generative adversarial network*. GAN is introduced in ref. [18] which was proposed to train a generative model. GAN consists of two blocks: a generative model ($G$) that learn the data distribution ($p(x)$), and a discriminative model ($D$) that estimates the probability that a sample came from the training data ($X$) rather than from the generator ($G$). These two models can be non-linear mapping functions such as two neural networks. To learn the generator distribution $p_g$ over data $x$, a differentiable mapping function is built by generator $G$ to map a prior noise distribution $p_z(z)$ to the data space as $G(z; \theta_g)$. The discriminator function $D(x; \theta_d)$ returns a single scalar that represents the probability of $x$ coming from the real data rather than from generator distribution $p_g$. The goal of the generator is to fool the discriminator, which tries to distinguish between true and generated data. Training of $D$ ensures that the discriminator can properly distinguish samples coming from both training samples and the generator. $G$ and $D$ are simultaneously trained to minimize $\log(1 - D(G(z)))$ for $G$ and maximize $\log(D(x))$ for $D$. It forms a two-player min–max game with value function $V(G, D)$

$$\min_G \max_D V(G, D) = E_{x \sim p_x(x)}[\log(D(x))] + E_{z \sim p_z(z)}[1 - \log(D(G(z)))] \quad (2)$$

*Locality sensitive hashing generative adversarial network*. For LSH-GAN, a subsampling of real data $p_{x_s}(x_s)$ is augmented with the prior noise distribution, $p_z(z)$. Due to this additional information in generator, we assume that the probability $D(G(\hat{z}))$ will increase by a factor, $\zeta$.

The value function of LSH-GAN can be written as:

$$L(D, G) = \min_G \max_D (E_{x \sim p_{data}(x)} \log(D(x)) + E_{z \sim p_{\hat{z}}(\hat{z})} \log(1 - D(G(\hat{z})))) \quad (3)$$

**Proposition 1.** *$L(D, G)$ is maximized with respect to discriminator ($D$), for a fixed generator ($G$), when*

$$D_G^*(x) = \frac{p_{data}(x)(1 - \zeta)}{p_{data}(x) + p_g(x)} \quad (4)$$

*Proof.* Equation (3) can be written as

$$L(D, G) = \int_x p_{data}(x) \log(D(x)) dx + \int_{\hat{z}} p_{\hat{z}}(\hat{z}) \log(1 - \{D(G(\hat{z})) + \zeta\}) d\hat{z}$$
$$= \int_x p_{data}(x) \log(D(x)) + p_g(x) \log(1 - \{D(x) + \zeta\}) dx \quad (5)$$

[As the range of $D(G(\hat{z}))$ is within the domain of real data $x$ so we can write this]

We know that, the function $y = a \log x + b \log(1 - (x + \zeta))$ will have maximum value, at $x = \frac{a(1 - \zeta)}{a + b}$, for any $(a, b) \in R^2\{0, 0\}$ and $\zeta \in (0, 1)$. So, the optimum value of $D$ for a fixed generator, $G$ is:

$$D_G^*(x) = \frac{p_{data}(x)(1 - \zeta)}{p_{data}(x) + p_g(x)} \quad (6)$$

The training objective for discriminator $D$ is to maximize the log-likelihood of the conditional probability $P(Y = y|x)$, where $Y$ signifies whether $x$ is coming from real

data distribution($y = 1$) or coming from the generator($y = 0$). Now the equation (3) can be written as

$$C(G) = \max_D L(G, D)$$
$$= \left( E_{x \sim p_{data}(x)} \log(D_G^*(x)) + E_{\hat{z} \sim p_g(\hat{z})} \log(1 - D_G^*(G(\hat{z}))) \right)$$
$$= (E_{x \sim p_{data}(x)} \log(D_G^*(x)) + E_{x \sim p_g(x)} \log(1 - D_G^*(x)))$$
$$= E_{x \sim p_{data}(x)} \log \frac{p_{data}(x)(1 - \zeta)}{p_{data}(x) + p_g(x)} + E_{x \sim p_g(x)} \log \left( 1 - \frac{p_{data}(x)(1 - \zeta)}{p_{data}(x) + p_g(x)} \right) \quad (7)$$

**Theorem 1.** *At $p_g(x) = p_{data}(x)$ (global minimum criterion of value function $L(G, D)$), the value of $C(G)$ is less than $(-\log 4)$.*

*Proof.* From equation (7) we get

$$C(G) = E_{x \sim p_{data}(x)} \log \left( \frac{p_{data}(x)(1 - \zeta)}{p_{data}(x) + p_g(x)} \right) + E_{x \sim p_g(x)} \log \left( 1 - \frac{p_{data}(x)(1 - \zeta)}{p_{data}(x) + p_g(x)} \right)$$
$$= E_{x \sim p_{data}(x)} \log \left( \frac{p_{data}(x)(1 - \zeta)}{p_{data}(x) + p_g(x)} \right) + E_{x \sim p_g(x)} \log \left( \frac{\zeta p_{data}(x) + p_g(x)}{p_{data}(x) + p_g(x)} \right)$$
$$= \left[ \log(1 - \zeta) + E_{x \sim p_{data}(x)} \log \left( \frac{p_{data}(x)}{p_{data}(x) + p_g(x)} \right) \right]$$
$$+ \left[ E_{x \sim p_g(x)} \log \left( 1 + \frac{\zeta p_{data}(x)}{p_g(x)} \right) + E_{x \sim p_g(x)} \log \left( \frac{p_g(x)}{p_{data}(x) + p_g(x)} \right) \right]$$
$$= \left[ \log(1 - \zeta) + E_{x \sim p_g(x)} \log \left( 1 + \frac{\zeta p_{data}(x)}{p_g(x)} \right) \right]$$
$$+ \left[ E_{x \sim p_{data}(x)} \log \left( \frac{p_{data}(x)}{p_{data}(x) + p_g(x)} \right) + E_{x \sim p_g(x)} \log \left( \frac{p_g(x)}{p_{data}(x) + p_g(x)} \right) \right]$$
$$= \left[ \log(1 - \zeta) + E_{x \sim p_g(x)} \log \left( 1 + \frac{\zeta p_{data}(x)}{p_g(x)} \right) \right]$$
$$+ \left[ (-\log 4) + 2 JSD(p_{data}(x) || p_g(x)) \right] \quad (8)$$

where, $JSD(p_{data}(x) || p_g(x))$ represents Jensen–Shannon divergence between two distributions $p_{data}$ and $p_g$. Now, if the two distributions are equal, Jensen–Shannon divergence (JSD) will be zero. Thus, for global minimum criterion of the value function ($p_g = p_{data}$) Eq. (8) reduces to

$$C(G) = \log(1 - \zeta) + \log(1 + \zeta) + (-\log 4) = \log \frac{(1 - \zeta^2)}{4} \leq (-\log 4) \quad (9)$$

This completes the proof.

As explained in the above proof, the generator function of LSH-GAN does not take prior noise ($p(z)$) as input instead a subsample of the original dataset is augmented with the noise. By adding this term, without loss of generality we assume that the probability $D(G(\hat{z}))$ will increase by a term $\zeta$. The assumption is justified by the input of the generator, which takes some amount of real data samples with the noise ($p(z)$) from the very first step of the training. The output of the generator $G(\hat{z})$, is more close to the real data distribution than the output $G(z)$ generated from the random noise ($p(z)$). This cause a small increase of the value $D(G(\hat{z}))$, which we assume as $\zeta$. This ultimately leads to a lower value of the cost function than $(-\log 4)$, which is the value of the cost function of original GAN at the global minimum criterion of value function $L(G, D)$ ($p_g(x) = p_{data}(x)$).

**Reporting summary**. Further information on research design is available in the Nature Research Reporting Summary linked to this article.

## Data availability
All the datasets are downloaded from Gene Expression Omnibus (GEO) https://www.ncbi.nlm.nih.gov/geo/, with accession no. GSE36552 (Yan Dataset), GSE65525 (Klein dataset), GSM1832359 (Pollen dataset), GSE67835 (Darmanis dataset), and GSE72056 (Melanoma dataset).

## Code availability
The corresponding software is available at https://github.com/Snehalikalall/LSH-GAN and https://doi.org/10.5281/zenodo.5903223.

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

## Acknowledgements

We would like to acknowledge support from J.C. Bose Fellowship [SB/S1/JCB-033/2016 to S.B.] by the DST, Govt. of India; SyMeC Project grant [BT/Med-II/NIBMG/SyMeC/2014/Vol. II] given to the Indian Statistical Institute by the Department of Biotechnology (DBT), Govt. of India; Inspire DST Project.

## Author contributions

S.L. and S.R. equally contributed to this work. S.B. supervised the whole work.

## Competing interests

The authors declare no competing interests.
