## [Peer Review File · Communications Biology]

Reviewers' comments:

Reviewer #1 (Remarks to the Author):

In this paper, the authors propose an improved version, LSH-GAN (generative adversarial network), to generate cell samples for the downstream analysis of scRNA-seq data. The main motivation of this paper is that the downstream analysis of scRNA-seq data is often limited due to the lack of enough cell samples. The manuscript is easy to follow. The method outperforms some old methods by a good margin. But the manuscript should be improved with addressing the following major issues:

1) Considering the publication year of GAN (2015), the comparison may consider more recent methods, e.g., f-gan or w-gan. Notably, the LSH step seems not exclusive to the other GAN variants. Is there any specific reason that the original GAN is chosen instead of more advanced variants, e.g., w-gan?

2) In Page 7, feature selection for cell clustering, it is unclear to me the feasibility of the cell clustering without generated samples. A more detailed comparison should be added.

3) Fig.3 used umap, but fig.4 used tsne. Could you give a reason for the different choices in visualization?

4) Theorem 1 mentions that the objective for LSH-GAN is less than the standard GAN. From my point of view, this is a correct result but does not provide any insight because the critical point is that the generator uses the data distribution, which naturally reduces the cross-entropy (which also explains the ζ term). More discussions on the theoretical results is encouraged.

5) From the biological view, there are several other questions. (1) what is the biological meaning of the generated samples? In natural images, fid is frequently used. Here, Wasserstein distance is used to evaluate the generated samples. Is there any evidence supporting the choice? (2) Does this method consider batch effect in data collection? Seurat indeed provides the de-batch method.

6) The hyperparameter analysis is weak. Why is t so small? Is that due to the data size and sampling size?

Some minor issues:

1. Boldfacing the best-performing results could improve the readability of the tables.

2. The figures are in pool layouts, e.g., there are overlappings and almost invisible legends in fig.2.

3. Proofreading is necessary. Typos and grammar mistakes, e.g.:

page 1: "Recently computational researchers gaining interest in this field...."

page 2: "LSH-GAN can able to generate realistic samples in a faster way than the traditional GAN..."

etc.

Reviewer #2 (Remarks to the Author):

In this manuscript, the authors proposed LSH-GAN, a generative model for producing new realistic single cell RNA-seq data. Combining GAN (generative adversarial network) with LSH (local sensitive hashing) is interesting. However, using GAN-based models for generating realistic in silico data (e.g.,

single cell RNA-seq) has been used in various previous works, such as cscGAN (<https://doi.org/10.1038/s41467-019-14018-z>) and scIGANs (<https://doi.org/10.1093/nar/gkaa506>). The differences between LSH-GAN and previous works are limited. Here are the specific comments.

Comments

- 1) The biological meaning of generated scRNA-seq data is not well illustrated. The authors should clarify how the generated scRNA-seq data could help better capture the differential expressed genes. For example, considering two cases, 1) give only original scRNA-seq data, and 2) give both original scRNA-seq data and generated data, then the authors should evaluate whether the case 2) could achieve a better differential expressed genes or clustering results.
- 2) The intuition of LSH sampling is not very clear. LSH sampling seems to be a fast way for finding nearest neighbors for high dimensional data. Please indicate why the LSH sampled data (neighbors) together with noise could help better generate the data. What if the LSH sampled data were replaced with the sample point itself ?
- 3) Please demonstrate whether LSH sampling strategy is general which can also contribute to a better generation in other datasets, such as biological datasets, or image datasets.
- 4) The comparison to scIGANs (<https://doi.org/10.1093/nar/gkaa506>) in all benchmark experiments.
- 5) The best results in the tables should be marked in bold.

Answer of Reviewer #1

In this paper, the authors propose an improved version, LSH-GAN (generative adversarial network), to generate cell samples for the downstream analysis of scRNA-seq data. The main motivation of this paper is that the downstream analysis of scRNA-seq data is often limited due to the lack of enough cell samples. The manuscript is easy to follow. The method outperforms some old methods by a good margin. But the manuscript should be improved with addressing the following major issues:

1) Considering the publication year of GAN (2015), the comparison may consider more recent methods, e.g., f-gan or w-gan. Notably, the LSH step seems not exclusive to the other GAN variants. Is there any specific reason that the original GAN is chosen instead of more advanced variants, e.g., w-gan?

Answer: There is no specific reason for choosing the original GAN as a comparative method. We choose this to show that incorporation of LSH-step with GAN performed better than the original GAN. As advised by the reviewer now we have compared LSH-GAN with more recent and advanced GAN variants such as f-gan and w-gan. Please see the table-2, figure-3F and subsection 'Gene selection in HDSS scRNA-seq data' (page-5) of the revised version of the manuscript.

2) In Page 7, feature selection for cell clustering, it is unclear to me the feasibility of the cell clustering without generated samples. A more detailed comparison should be added.

Answer: Please note that we utilized generated samples for feature (gene) selection. Here our aim is to see whether the selected features from the generated data are effective for cell clustering. We have rewrite the subsection to avoid any confusion. Please see the subsection 'Selected genes using LSH-GAN can effectively predict cell clusters' of the revised version of the manuscript.

3) Fig.3 used umap, but fig.4 used tsne. Could you give a reason for the different choices in visualization?

Answer: For visualization (creating 2 dimensional embedding) we always follow one method (here UMAP). Figure-4 is also created using UMAP, however the caption is wrong. We have now corrected the caption in the revised version.

4) Theorem 1 mentions that the objective for LSH-GAN is less than the standard GAN. From my point of view, this is a correct result but does not provide any insight because the critical point is that the generator uses the data distribution, which naturally reduces the cross-entropy (which also explains the ζ term). More discussions on the theoretical results is encouraged.

Answer: We agreed with the reviewer that cross entropy is reduced due to ζ term.

By including sub-sample of original data with prior noise, we assume that the probability $D(G(z'))$ will increase by a factor ζ . We later prove that the ζ term affects the cost function ($C(G)$) of the generator (equation 5 and 7), and finally $C(G)$ is proved to be less than $(-\log 4)$, when $p_g = p_{data}$.

As advised by the reviewer we have now included more discussions of the theoretical results in the revised version. Please see the last paragraph of the proof (page-11) of the revised version of the manuscript.

5) From the biological view, there are several other questions. (1) what is the biological meaning of the generated samples? In natural images, fid is frequently used. Here, Wasserstein distance is used to evaluate the generated samples. Is there any evidence supporting the choice? (2) Does this method consider the batch effect in data collection? Seurat indeed provides the de-batch method.

Answer: To know the biological significance we have utilized several gene selection methods to select genes from the generated samples. This is further evaluated using single cell clustering techniques. Please see table-3 and subsection 'Gene selection from generated sample of HDSS scRNA-seq data' for details.

Wasserstein distance is a common way to measure the distances between two data distributions. The Wasserstein metric has major advantages over KL divergence and other measures. For example, the Wasserstein metric does not require both measures to be on the same probability space, whereas KL divergence requires both measures to be defined on the same probability space.

As advised by the reviewer we have now included one analysis to know how LSH-GAN is affected by the data of different batches. For this we have downloaded one dataset from [Tran et al. A benchmark of batch-effect correction methods for single-cell RNA sequencing data (Genome Biol 21, 12 (2020))], consisting of several batches and applied LSH-GAN on it. We have now included a subsection 'LSH-GAN is robust for data with different batches' (page-7) of the revised version of the manuscript to explain the analysis and the results.

6) The hyperparameter analysis is weak. Why is t so small? Is that due to the data size and sampling size?

Answer: The parameter t controls the amount of sampling. It represented the number of passes of the sampling technique. In each pass the data is subsampled sequentially. The first pass takes the original data as input and subsample it using a fixed k (Number of nearest neighbor). In the next pass the sampled data is taken as input, and subsequently the process goes on. The choice of ' t ' depends on the sample size of the original data and the amount of sampling we require. We explore different choices of ' t ' and ' k ' on five publicly available datasets and choose the parameters which give best results. Please see supplementary table-1 for the results.

Some minor issues:

1. Boldfacing the best-performing results could improve the readability of the tables.

Answer: As suggested we have marked the best results in bold font.

2. The figures are in pool layouts, e.g., there are overlappings and almost invisible legends in fig.2.

Answer: As suggested by the reviewer, the figure 2 is corrected.

3. Proofreading is necessary. Typos and grammar mistakes, e.g.:

page 1: "Recently computational researchers gaining interest in this field...."

page 2: "LSH-GAN can able to generate realistic samples in a faster way than the traditional GAN..."

etc.

Answer: As suggested we have corrected the grammatical mistakes.

Answer of Reviewer #2

In this manuscript, the authors proposed LSH-GAN, a generative model for producing new realistic single cell RNA-seq data. Combining GAN (generative adversarial network) with LSH (local sensitive hashing) is interesting. However, using GAN-based models for generating realistic in silico data (e.g., single cell RNA-seq) has been used in various previous works, such as cscGAN (<https://doi.org/10.1038/s41467-019-14018-z>) and scIGANs (<https://doi.org/10.1093/nar/gkaa506>). The differences between LSH-GAN and previous works are limited. Here are the specific comments.

Comments

1) The biological meaning of generated scRNA-seq data is not well illustrated. The authors should clarify how the generated scRNA-seq data could help better capture the differential expressed genes. For example, considering two cases, 1) give only original scRNA-seq data, and 2) give both original scRNA-seq data and generated data, then the authors should evaluate whether the case 2) could achieve a better differential expressed genes or clustering results.

Answer: As advised by the reviewer we have now performed the experiment to know generated scRNA-seq data could help better capture the differential expressed genes. For finding out DE

genes we have utilized five popular gene selection methods (GLM_PCA, HVG of Seurat, Fano factor, CV² index, and M3Drop) of scRNA-seq data analysis domain. Given only (case-1): original scRNA-seq data and (case-2): both original and generated data, we have found DE genes using the five methods. Table-3 represents the clustering result (ARI values) for the case-1 and case-2. The last column of the table-3 represents the clustering results on the original scRNA-seq data (case-1). Please see table-3 and the subsection 'Gene selection in HDSS scRNA-seq data' for the detailed analysis.

2) The intuition of LSH sampling is not very clear. LSH sampling seems to be a fast way for finding nearest neighbors for high dimensional data. Please indicate why the LSH sampled data (neighbors) together with noise could help better generate the data. What if the LSH sampled data were replaced with the sample point itself ?

Answer: The rationale behind this may be that, LSH sampling step captures non-redundant and widely separated samples from original datasets which helps to learn the complex distribution of the data in a holistic way. Please note that the intuition of LSH step is to provide a prior sense of information about real data distribution to the generator network. We have provided a theoretical proof to explain the rationale of this data augmentation process before giving to the generator of GAN. We have now mentioned this in the subsection 'LSH-step' (page-2) of the revised version of the manuscript.

Please note that, we retain some original sample points from the original scRNA-seq data using LSH based sampling. These sample points are augmented with random noise and given as input to the generator for training.

3) Please demonstrate whether LSH sampling strategy is general which can also contribute to a better generation in other datasets, such as biological datasets, or image datasets.

Answer: We have demonstrated the LSH sampling strategy in biological dataset (scRNA-seq data). As advised by the reviewer we have now applied LSH-GAN on a separate biological dataset (bulk RNA-seq data). We downloaded the bulk RNA-seq cortex data of WT (wild type) and AD (Alzheimer's disease) phenotype mice, which are known to form amyloid-beta (A β) plaques, at three different ages (2M, 4M, and 7M) from a GSE104775 dataset released to the NCBI GEO website. We downloaded the processed data from Jinhee Park et al. (A practical application of generative adversarial networks for RNA-seq analysis to predict the molecular progress of Alzheimer's disease, PloS Comp. Bio, 2020), which consists of gene expression values of 1,208 DEGs over 36 samples. We utilized a random forest classifier for discriminating the real and generated samples by LSH-GAN. The classifier gives a chance level performance (AUC score 0.57 \pm 0.03 for 10 fold cv) for discriminating between real and generated samples, which suggests the generated samples are highly similar with real samples.

As our main aim is to generate expression data from small sample single-cell datasets, we have mentioned the efficacy of LSH-GAN in other datasets in the Discussion section (page-8). The detailed analysis is given in the sec-4 of the supplementary text. Please see the fourth paragraph (page-8) of the Discussion section of the revised manuscript.

4) The comparison to scIGANs (<https://doi.org/10.1093/nar/gkaa506>) in all benchmark experiments.

Answer: Please note that the main purpose of scIGAN is single cell imputation. scIGANs converts the expression profiles of individual cells to images and feeds them to generative adversarial networks (GAN). The trained model is utilized to produce expression profiles which represent the realistic cells of defined types. The generated cells, rather than the observed cells, are then used to impute the dropouts of the real cells.

Please note that the objective of LSH-GAN is different from the objective of scIGAN. LSH-GAN is formulated for small sample scRNA-seq data generation, while the primary aim of scIGAN is data imputation. So, the scope for comparison between these two methods is limited. However, before imputation scIGAN first utilized the original version of GAN for data generation. Please note that we have compared LSH-GAN with original GAN and the other variants of GAN (such as f-GAN, w-GAN). Please see table-3 for detailed results.

5) The best results in the tables should be marked in bold.

Answer: As suggested we have marked the best results in bold font.

Reviewers' comments:

Reviewer #2 (Remarks to the Author):

Most of my concerns were addressed by the authors. Here are some minor comments.

- 1) Only ARI was used as a metric for evaluating clustering, could other clustering metrics, such as NMI also be reported together with ARI?
- 2) From table1, the difference between real data and generated data could be measured by the Wasserstein distance. Could this metric also be reported?
- 3) Applying generative models (e.g., GAN) to single cell analysis is a very active topic. Many efforts have been dedicated to this. Some very related references should be mentioned and acknowledged (Proc Natl Acad Sci USA, 118(15): e2101344118, 2021; Nat Mach Intell, 3, 536–544 (2021); Nucleic Acids Res, 2020, 48(15): e85-e85.)

(Comments regarding Reviewer #1's concerns)

Major concerns:

Q1: Addressed satisfactorily.

Q2: Only rewrite a sentence is not enough. It seems that the original question is whether selected genes can be used for cell clustering in some real data instead of generated data. I think the authors should design an experiment to verify whether the selected features (genes) from generated samples can be used for achieving a better clustering performance in real scRNA-seq datasets.

Q3: Addressed satisfactorily.

Q4: Addressed satisfactorily.

Q5: Addressed satisfactorily.

Q6: The hyperparameter analysis is still weak, but this is not critical.

Minor concerns:

Addressed satisfactorily.

A new minor concern:

Applying generative models (e.g., GAN) to single cell analysis is a very active topic. Many efforts have been dedicated to this. Some very related references should be mentioned and acknowledged (Nat Mach Intell 3, 536–544 (2021); Nucleic Acids Res, 2020, 48(15): e85-e85.) It is expected that the authors give a brief discussion about this direction and cite these papers.

Answer of Reviewer 2 for review 1

Q2: Only rewrite a sentence is not enough. It seems that the original question is whether selected genes can be used for cell clustering in some real data instead of generated data. I think the authors should design an experiment to verify whether the selected features (genes) from generated samples can be used for achieving a better clustering performance in real scRNA-seq datasets.

Answer: Please note that the selected features from the combined data (original with generated data) are utilized to cluster the real sc-RNAseq data. So, we utilize the augmented data for feature selection only, which is further used to cluster the real scRNA-seq data. We have now updated the caption and some part of Table-3 to avoid any confusion.

Minor concerns:

Addressed satisfactorily.

A new minor concern:

Applying generative models (e.g., GAN) to single cell analysis is a very active topic. Many efforts have been dedicated to this. Some very related references should be mentioned and acknowledged (Nat Mach Intell 3, 536–544 (2021); Nucleic Acids Res, 2020, 48(15): e85-e85.) It is expected that the authors give a brief discussion about this direction and cite these papers.

Answer: As advised the references are now cited in references 18 and 19.

Answer of reviewer 2

Most of my concerns were addressed by the authors. Here are some minor comments.

1) Only ARI was used as a metric for evaluating clustering, could other clustering metrics, such as NMI also be reported together with ARI?

Answer: As advised we have now updated the table-3 with the NMI score. Please see Table-3 of the revised version of the manuscript.

2) From table1, the difference between real data and generated data could be measured by the Wasserstein distance. Could this metric also be reported?

Answer: The Wasserstein metric is reported in table-1. Please see Table-1 of the revised version of the manuscript.

3) Applying generative models (e.g., GAN) to single cell analysis is a very active topic. Many efforts have been dedicated to this. Some very related references should be mentioned and acknowledged (Proc Natl Acad Sci USA, 118(15): e2101344118, 2021; Nat Mach Intell, 3, 536–544 (2021); Nucleic Acids Res, 2020, 48(15): e85-e85.)

Answer: As advised the references are now cited in references 18, 19 and 20.

REVIEWERS' COMMENTS:

Reviewer #2 (Remarks to the Author):

All questions have been addressed. I have no more comments.